# Neutralizing Antibodies against the SARS-CoV-2 Ancestral Strain and Omicron BA.1 Subvariant in Dogs and Cats in Mexico

**DOI:** 10.3390/pathogens12060835

**Published:** 2023-06-16

**Authors:** Freddy Dehesa-Canseco, Roxana Pastrana-Unzueta, Nadia Carrillo-Guzmán, Francisco Liljehult-Fuentes, Juan Diego Pérez-De la Rosa, Humberto Ramírez-Mendoza, Jose Guillermo Estrada-Franco, Roberto Navarro-López, Jesús Hernández, Mario Solís-Hernández

**Affiliations:** 1Comisión México-Estados Unidos para la Prevención de la Fiebre Aftosa y otras Enfermedades Exóticas de los Animales (CPA), Servicio Nacional de Sanidad, Inocuidad y Calidad Agroalimentaria (SENASICA), Secretaría de Agricultura y Desarrollo Rural (SADER), Ciudad de Mexico 05110, Mexico; freddy_1994333@hotmail.com (F.D.-C.); unzuetaqfb@gmail.com (R.P.-U.); nadia.carrillo@senasica.gob.mx (N.C.-G.); francisco.liljehult@senasica.gob.mx (F.L.-F.); roberto.navarro@senasica.gob.mx (R.N.-L.); 2Centro Nacional de Servicios de Constatación en Salud Animal (CENAPA), Servicio Nacional de Sanidad, Inocuidad y Calidad Agroalimentaria (SENASICA), Secretaría de Agricultura y Desarrollo Rural (SADER), Cuernavaca 62574, Mexico; juan.perez@senasica.gob.mx; 3Departamento de Microbiología e Inmunología, Facultad de Medicina Veterinaria y Zootecnia (FMVZ), Universidad Nacional Autónoma de México (UNAM), Ciudad de Mexico 04510, Mexico; betosram@comunidad.unam.mx; 4Centro de Biotecnología Genómica, Instituto Politécnico Nacional, Reynosa 88710, Mexico; joseestradaf@hotmail.com; 5Laboratorio de Inmunología, Centro de Investigación en Alimentación y Desarrollo, A. C. (CIAD), Hermosillo Sonora 83304, Mexico; jhdez@ciad.mx

**Keywords:** cats, COVID-19, dogs, microneutralization, PRNT-90, SARS-CoV-2, seroprevalence

## Abstract

SARS-CoV-2 mainly affects humans; however, it is important to monitor the infection of companion and wild animals as possible reservoirs of this virus. In this sense, seroprevalence studies in companion animals, such as dogs and cats, provide important information about the epidemiology of SARS-CoV-2. This study aimed to evaluate the seroprevalence of neutralizing antibodies (nAbs) against the ancestral strain and the Omicron BA.1 subvariant in dogs and cats in Mexico. Six hundred and two samples were obtained from dogs (n = 574) and cats (n = 28). These samples were collected from the end of 2020 to December 2021 from different regions of Mexico. The presence of nAbs was evaluated using a plaque reduction neutralization test (PRNT) and microneutralization (MN) assays. The results showed that 14.2% of cats and 1.5% of dogs presented nAbs against the ancestral strain of SARS-CoV-2. The analysis of nAbs against Omicron BA.1 in cats showed the same percentage of positive animals but a reduced titer. In dogs, 1.2% showed nAbs against Omicron BA.1. These results indicate that nAbs were more frequent in cats than in dogs and that these nAbs have a lower capacity to neutralize the subvariant Omicron BA.1.

## 1. Introduction

Coronavirus disease 2019 (COVID-19), caused by severe acute respiratory syndrome coronavirus 2 (SARS-CoV-2), was first reported in December 2019 in the city of Wuhan, Hubei Province, central China and was associated with a significant number of cases of pneumonia of unknown origin [1]. Since then, the virus has spread to different countries, causing more than seven hundred million infections and more than six million deaths [1,2]. In Mexico, the first case reported was in February 2020, and by 31 May 2023, 7.6 million cases had been reported [3], and globally, 767 million cases [2].

In addition to infecting humans, SARS-CoV-2 can infect domestic and zoo animals. However, the role of such an infection in transmitting the virus or as a reservoir of the disease remains unknown. The first case of anthropozoonosis was reported in Hong Kong, involving the dog of a patient who presented with COVID-19 diagnosed by RT–qPCR [4]. Since then, several reports of infected companion animals have been reported in different parts of the world [5,6,7,8,9,10,11,12,13,14,15,16]. In Mexico, the first case of SARS-CoV-2 in dogs was reported by the National Service of Agri-Food Health, Safety and Quality (SENASICA, for its Spanish acronym) in April 2020. In this case, four dogs were in contact with RT–qPCR-positive humans and one dog was found to be positive for SARS-CoV-2 by RT–qPCR [17].

The emergence of variants of SARS-CoV-2 has provoked reinfections and reduced neutralizing responses to vaccines or infected individuals, especially with Omicron subvariants [18]. The Omicron BA.1 was the first subvariant of the Omicron family. The virus has continued evolving, and new subvariants such as BA.5 have emerged. However, the recent family of subvariants, BQ and XBB, have shown a significant reduction in the neutralizing response of previous infections, even after repeated immunizations [19]. In this regard, there is limited information on the response of cats and dogs to Omicron subvariants. Recent reports have shown that cats and dogs are susceptible to infection with the Omicron BA.1 subvariant, which seems less pathogenic than infections with the D614G and Delta variants [20,21,22]. Transmission of SARS-CoV-2 from humans to dogs and cats is well-documented, and most infections result from very close contact with their COVID-19-infected owners. However, the impact of these new subvariants on the clinical manifestations and duration of disease in pets or transmissibility between humans and pets remains unknown [23,24,25].

Because SARS-CoV-2 can infect animals, it is essential to determine the extent of the disease in these populations via comprehensive analysis within the context of One Health. Therefore, this study aimed to evaluate the seroprevalence of neutralizing antibodies against the ancestral strain and Omicron BA.1 subvariants in dogs and cats from different regions of Mexico collected from the end of 2020 to December 2021.

## 2. Materials and Methods

### 2.1. Serum Samples

A total of 1111 serum samples were included in this study, of which 1052 were from dogs, and 59 were from cats. Serum samples were collected from 25 states: Guerrero, Coahuila, Chihuahua, Durango, Michoacán, Nuevo León, Yucatán, Jalisco, México City, Estado de México, Oaxaca, Puebla, Quintana Roo, San Luis Potosí, Sinaloa, Tabasco, Tamaulipas, Yucatán, Zacatecas, Aguascalientes, Baja California, Baja California Sur, Campeche, Colima, and Chiapas, derived from seroepidemiological surveillance of SARS-CoV-2 conducted for SENASICA. In addition, this study included five sources of cats and dogs (Table 1): (1) Samples from animals (household) that were in contact with people positive for the virus by RT–qPCR (Thermo Fisher Scientific^®^, San Hose, CA, USA). (2) Samples from urban free-ranging dogs and cats were collected during a surgical sterilization program organized by the National Center for Preventive Programs and Disease Control and SENASICA. (3) Samples from animals living in shelters (low contact with humans, but high contact with other dogs). (4) Samples from dogs of the federal border inspection agencies used mainly for drug inspection duties (low contact with other dogs, but high contact with humans).

Dogs and cats were classified according to their health condition described by the owners in the medical records or identified at the time of sampling: 1, mild (nasal congestion, fatigue, cough); 2, moderately ill (gastroenteritis, nasal congestion, fatigue, cough, dyspnea, pyrexia, emesis, and pulsating chest pain); 3, severe disease (bilateral pneumonia, acute respiratory distress, and alterations in lung function); and 4, healthy (asymptomatic). This classification was performed according to previous reports [26,27].

Blood samples were collected from the cephalic or jugular vein (approximately 3–5 mL) using Vacutainer tubes without an anticoagulant but with a separating gel, centrifuged to obtain serum, and stored at −20 °C until use. The samples were collected from the end of 2020 to December 2021. The samples were processed at the biosafety diagnostic laboratory level 3 (LBS3) of the Mexico–United States Commission for the Prevention of Foot-and-Mouth Disease and Other Exotic Animal Diseases (CPA) of the National Service for Disease Control, Safety, and Agri-food Quality (SENASICA).

This work agreed with the general ethical principles and guidelines established by the Official Mexican Standard NOM-062-ZOO-1999. Additionally, the Internal Committee approved the protocol for the Care and Use of Laboratory Animals (CICUAL) belonging to the CPA-SENASICA (CICUAL-CPA-001-2022).

### 2.2. Cell Line

Vero C1008 (Vero 76, clone E6, Vero E6) cells (ATCC-CRL-1586 cells) were maintained in Dulbecco’s modified Eagle’s medium (DMEM) (Gibco^®^, Grand Island, NY, USA, Cat. No. 11995-065) with 5% fetal bovine serum (FBS) (ATCC^®^, Manasass, VA, USA, Cat. No. 30-2020). The serum was subjected to gamma radiation with cobalt 60 and inactivated at 56 °C for one hour. The medium was supplemented with a 2% penicillin–streptomycin–amphotericin B suspension (Sigma^®^, Colobo, TX, USA, Cat. No. A5955).

### 2.3. Virus

In this study, the viruses hCoV-19/Mexico/CPALB32021036/2020 and hCoV-19/Mexico/CPALBS32021032/2022 were used to evaluate the neutralizing capacity of serum samples by MN and PRNT90. The viruses were isolated and characterized in the facilities of the LBS3 of CPA/SENASICA. The viruses showed 99.9% identity with the reference sequence Wuhan-Hu-1 (GenBank: NC_045512.2) and 99.9% identity with the sequence SARS-CoV-2/human/USA/CA-CDC-STM-B8VEK3CPH/2022 (GenBank: OM464899.1), identified as the Omicron (BA.1) subvariant.

### 2.4. RT–qPCR

Real-time RT–qPCR was based on detecting SARS-CoV-2 regions using primers N1, N2, and RP (Appendix A) (2019-nCoVPC and RP). A SARS-CoV-2 (2019-nCoV) CDC qPCR Probe Assay and Detection kit for the 2019 novel coronavirus (2019-nCoV RNA (PCR-fluorescence probing) with a 7500 Fast Real-Time PCR System (Thermo Fisher Scientific^®^, San Jose, CA, USA) and CFX96 Touch Real-Time PCR Detection System (Bio-Rad Laboratories^®^, Hercules, CA, USA) were used.

### 2.5. Plaque Reduction Neutralization Test (PRNT90)

Plaque reduction neutralization was conducted as previously described [28] with some modifications. Six-well cell culture plates with flat bottoms (NEST^®^, Wuxi, China, cat. No. 07-6006) were used, and Vero C1008 cells were cultured at a concentration of 1.5 × 10^6^ cells/well for 12–18 h before performing the test to reach 95–100% confluence. Serum samples were heat-inactivated at 56 °C for 60 min. Serial dilutions (1:10 to 1:5120) were prepared in DMEM (Gibco^®^, Cat. No. 11995-065) supplemented with 2% FBS and a 2% penicillin–streptomycin–amphotericin B suspension.

Dilution of the working virus was adjusted to 30–40 plaque-forming units (PFUs) per well, corresponding to 10^4^ dilutions of the seed virus. The active virus was added (15–20 final PFU) to each serum sample and incubated at 37 °C and 5% CO_2_ for one hour. Additionally, corresponding dilutions were performed to verify the viral titer and interpret the PRNT90 results in each experiment. For this purpose, 10 to 10^5^ dilutions of the virus were used. After incubation, the maintenance medium was removed from six-well plates previously seeded with Vero C1008 cells, and 200 µL of the virus–serum mixture was added. Similarly, 100 µL of the dilutions corresponding to the virus dilution were added to verify the viral titer, and the plates were incubated at 37 °C and 5% CO_2_ for 60 min. Subsequently, 2 mL of 1.6% LE agarose and 8% MEM 2× (Gibco^®^, Cat. No. 11995-065), FBS plus 2% antibiotic was added to each well, solidified, and incubated at 37 °C and 5% CO_2_ for 48 h. Finally, 1 mL of Neutral Red Solution (Sigma–Aldrich, Gillingham, UK, Cat. No. N2889) diluted to 0.058% in DPBS (Gibco^®^, Cat. No. 11995-065) was added to each well. The plates were incubated at 37 °C with 5% CO_2_ for 4–6 h. Subsequently, the neutral red solution was removed, and the measurement and interpretation of the results were analyzed. Samples were considered positive when a serum dilution of at least 1:10 reduced SARS-CoV-2 lytic plaque formation by no less than 90% compared with the control [28].

### 2.6. Microneutralization

Microneutralization was conducted as previously described [29,30] but with some modifications. Vero C1008 cells were cultured at a concentration of 1.5 × 10^5^ cells/mL (100 μL/well) in 96-well cell culture plates; to reach 70–80% confluence, the cells were prepared 12–18 h in advance. Subsequently, an initial 1:10 dilution was performed with sera (previously inactivated) in 96-well plates with flat bottoms (Corning^®^, Coring, NY, USA cat. No. CLS3585-50EA). The solution consisted of DMEM (Gibco^®^, Cat. No. 11995-065) supplemented with 2% FBS (ATCC^®^, Cat. No. 30-2020) and 2% penicillin–streptomycin–amphotericin B suspension (Sigma^®^, Cat. No. A5955). Twofold serial dilutions of serum samples (1:10 to 1:5120) were prepared in 50 µL and 50 µL of the virus (100 TCID_50_) and incubated at 37 °C in 5% CO_2_ for 60 min. After incubation, 100 µL of the serum–virus mixture was transferred to 96-well plates with Vero cells, 100 µL of DMEM (Gibco^®^, Cat. No. 11995-065) with 2% FBS (ATCC^®^, Cat. No. 30-2020) was added, and the cells were incubated at 37 °C and 5% CO_2_ for 72 h. Samples were considered positive when a serum dilution of at least 1:10 did not neutralize [29,30,31].

### 2.7. Positive and Negative Controls

Positive controls for the PRNT90 and MN tests were obtained from guinea pigs (*Cavia porcellus*). The ancestral strain (hCoV-19/Mexico/CPALB32021036/2020) and Omicron BA.1 (hCoV-19/Mexico/CPALBS32021032/2022) were inactivated with cobalt-60 gamma rays (^60^Co), emulsified with Montanide ISA™ 51, and used to immunize guinea pigs (twice) (Appendix A). As a negative control, we used the serum of a dog negative for RT–qPCR and antibodies against SARS-CoV-2 (ID Screen^®^ SARS-CoV-2 Double Antigen Multispecies ELISA Kit).

### 2.8. Statistical Analysis

The correlation between MN and PRNT90 was determined using the Pearson correlation coefficient with the statistical program IBM^®^ SPSS^®^ Statistics version 25 and GraphPad Prism version 8. In all cases, a value of *p* < 0.05 was considered significant. The comparison between nAbs against the ancestral and Omicron BA.1 was determined by two-tailed Wilcoxon matched-pairs signed-ranks tests, and p values less than 0.05 were considered statistically significant. These analyses were performed with GraphPad Prism version 8.

## 3. Results

This study included blood samples of 28 cats and 574 dogs obtained from several states of Mexico (Table 1 and Figure 1). The states with positive samples were Puebla, Estado de Mexico, Mexico City, Chiapas, and Quintana Roo (Figure 1).

The household group included 273 samples of animals that lived in close contact with persons: 264 corresponded to dogs and 9 to cats. One hundred and thirteen samples were tested by RT–qPCR, and only three were positive (ID-545: gene N1, Ct 37.51, and gene N2, Ct 38.28; ID-549: gene N1, Ct 33.44; gene N2, Ct 37.46; ID-546: gene N1, Ct 30.14, and gene N2, Ct 32.01). The criteria for performing an RT–qPCR test was that the owner had a positive RT–qPCR test and that the animal presented at least one symptom of COVID-19. Under this criterion, 160 animal samples were not analyzed using RT–qPCR. However, all samples were analyzed using PRNT90 and MN. The results indicated that in this group, five samples showed nAbs against the ancestral strain of SARS-CoV-2: three cats (33.3%) and two dogs (0.73%) (Table 2 and Table 3). For nAbs against Omicron BA.1, three cats were positive, but sera from dogs were negative. Of note, titers against the ancestral strain in cats were higher than those against the Omicron BA.1 sublineage with both tests, PRNT90 (*p* = 0.0078) and MN (*p* = 0.0039).

The three animals with RT–qPCR-positive results had neutralizing antibodies (Table 2 and Table 3); one dog and one cat from the household group (ID-545 and ID-546, respectively) did not present with any clinical signs. However, one dog from the household group (ID-549) showed acute respiratory distress and pulmonary function alterations three weeks before sampling (without confirming the presence of bilateral pneumonia). This dog was in severe condition days after sample collection and was euthanized.

The second group comprised 155 samples from free-ranging urban dogs (n = 142) and cats (n = 13). RT–qPCR was not performed because none of the animals showed disease symptoms. The analysis of these samples revealed two dogs (1.4%) with nAbs against the ancestral strain (Table 3) and the Omicron BA.1 subvariant; however, one dog was negative for Omicron BA.1 with the MN test. None of the cats in this group showed nAbs against either strain.

The third group consisted of samples obtained from shelters (n = 124): 118 dogs and 6 cats. None of these samples were analyzed by RT–qPCR because there were no disease symptoms at the time of sample collection. The results showed that one dog (0.84%) and one cat (16.6%) were positive for nAbs against the ancestral strain and Omicron BA.1; however, the dog was negative for Omicron BA.1 with the MN test.

The fourth group included 50 samples from dogs of federal border inspection agencies. In this case, RT–qPCR was not performed. Four samples (8%) were positive for nAbs against the ancestral strain and the Omicron BA.1 subvariant.

We observed that antibody titers were higher when quantified with the PRNT90 assay than with MN, regardless of the ancestral strain or Omicron BA.1 subvariant. The most elevated titers observed in this study were 1:640 with PRNT90 and 1:320 with MN against the ancestral strain but were the lowest (1:320 and 1:160 for PRNT90 for MN, respectively) when using Omicron BA.1. Some cats and dogs with low titers against the ancestral strain were negative for Omicron BA.1., moreover, presented up to three logarithms of difference compared to the ancestral strain, especially with the MN test. However, even with this difference in sensitivity, both tests showed a strong correlation (r^2^ = 0.8109; *p* < 0.0001) (Figure 2). These results suggest that the PRNT90 technique is more sensitive than the MN test, even though the MN test is simpler to perform. Additionally, the results indicate that nAbs in cats and dogs could neutralize Omicron BA.1 but with lower potency.

The correlation between both techniques using the ancestral strain and Omicron BA.1 subvariant was r^2^ = 0.9878 (*p* < 0.0001) for PRNT90 and r^2^ = 0.8422 (*p* < 0.0001) for MN, indicating a strong correlation between the results of both techniques. These results suggest that despite a significant log difference between the results, there was no significant difference, and the results were closely related to both viruses.

## 4. Discussion

This study aimed to evaluate the seroprevalence of nAbs against the ancestral strain and Omicron BA., 1 subvariant of SARS-CoV-2, using PRNT90 and MN. For this purpose, 602 samples were analyzed, 574 corresponding to dogs and 28 to cats. Overall, our results demonstrated that the seroprevalence of nAbs against the ancestral strain was 14.2% in cats and 1.5% in dogs. The prevalence of nAbs against the Omicron BA.1 subvariant was similar to that of the ancestral variant but with lower titers in cats and 1.2% in dogs. The seroprevalence found in this study was similar to that reported in other countries [14,31,32,33,34,35,36]. In the case of dogs, most reports concord that seroprevalence is lower than that in cats, but there are studies with a diverse percentage of seroprevalence. The differences in the assays used to evaluate the antibodies could explain these differences because some cross-reactivity with other coronaviruses affecting dogs has been described.

In the present study, we confirmed the presence of infected dogs by RT–qPCR in two dogs and one cat from a household; the three animals were positive for nAbs. The animals were in contact with at least one person who tested positive for SARS-CoV-2. These results confirmed that dog owners are the most frequent sources of infection, as previously reported by others [37]. However, this is not the first report of dog infection in Mexico because SENASICA previously reported another case to the OMSA [33]. Interestingly, other dogs in contact with infected persons were negative for RT–qPCR but positive for antibodies [14,38,39]. One explanation is that SARS-CoV-2 has a low replication rate in dogs or that the infection occurred weeks before sample collection.

It has been reported that SARS-CoV-2 can infect dogs, felines, and wildlife animals [7,9,12,13,14,40,41,42]. Different studies have reported using PRNT90 and MN to evaluate the prevalence of nAbs against SARS-CoV-2 in dogs and cats. In Italy, neutralizing antibodies against SARS-CoV-2 were detected in 15 of 451 samples from dogs (3.3%) and 11 of 191 samples from cats (5.8%), with titers ranging from 1:20 to 1:160 and 1:20 to 1:1280, respectively. Positive samples were found in pets from households with people who were positive for SARS-CoV-2 [14]. In another study conducted in Brazil, the percentage of positive samples was low, at only 2% (2 of 96); based on the PRNT90 technique, one feline and one canine had titers of 1:80 and 1:40, respectively [33]. Other studies using ELISA have found prevalence rates ranging from 1.7% in cats to 0.8% in dogs [21,35,43,44]. These studies contribute to understanding the role of pets and other animals in the current pandemic.

In this study, we found that cats had higher antibody titers than dogs, consistent with previous reports [14,31,33,35]. The seroprevalence observed in this study was similar to those reported by other studies carried out in China [31], Thailand [36], Brazil [32,33], France [34], and Italy [14]. Overall, the results confirm that felines are more susceptible than dogs. Previous reports indicate that many RT–qPCR-positive dogs do not undergo seroconversion, leading to lower susceptibility to the virus [27]. To date, the role of felines in disseminating SARS-CoV-2 is unknown; therefore, it is important to establish seroepidemiological studies that involve a larger number of animals and other susceptible species that are companion animals, such as ferrets and hamsters.

The nAbs were analyzed using PRNT90 and MN techniques, widely employed to detect antibodies against SARS-CoV-2 in humans [28,29,30,45]. Different studies have proposed PRNT90 as the gold standard for confirming the presence of antibodies against SARS-CoV-2 [30,45]. However, this methodology has more significant technical difficulties. For example, it requires highly trained personnel to conduct the test; it also has higher costs, and fewer samples can be processed each time. In contrast, MN has lower technical difficulty and fees, and many samples can be processed simultaneously. The PRNT90 and MN utilized in the present study showed results similar to previous reports [14,32,33]. An interesting observation of this study was the positive correlation between PRNT90 and MN.

A limitation of this study was that the RT–qPCR results for all samples were unavailable, with the inability to establish a correlation between positive animals by RT–qPCR and PRNT90/MN. In those samples obtained from animals in households with humans positive for SARS-CoV-2, 113 samples were analyzed by RT–qPCR, and all contained antibodies. Nonetheless, it was impossible to perform RT–qPCR for the samples obtained from shelters, urban free-ranging animals, or federal border inspection agencies (n = 329). Another limitation of this study was that we could not confirm the virus variant infecting cats and dogs; however, based on the sampling dates, the infection was not caused by the Omicron BA.1 subvariant since it was not in circulation in Mexico at the time. Another limitation of this study was the lack of certified and approved reference strains and referenced serum in negative and positive controls. However, this study isolated and sequenced the viruses and deposited them in the GenBank. Additionally, with these viruses (ancestral and BA.1), we prepared negative and positive controls using non-infected and infected guinea pigs.

## 5. Conclusions

In this study, we used PRNT90 and MN assays to determine that cats in Mexico showed a higher seroprevalence than dogs, confirming their increased susceptibility to SARS-CoV-2 infection. Additionally, this study showed the presence of nAbs against Omicron BA.1, indicating that the titer against this variant in cats and dogs is lower than that against the ancestral variant.

## Figures and Tables

**Figure 1 pathogens-12-00835-f001:**
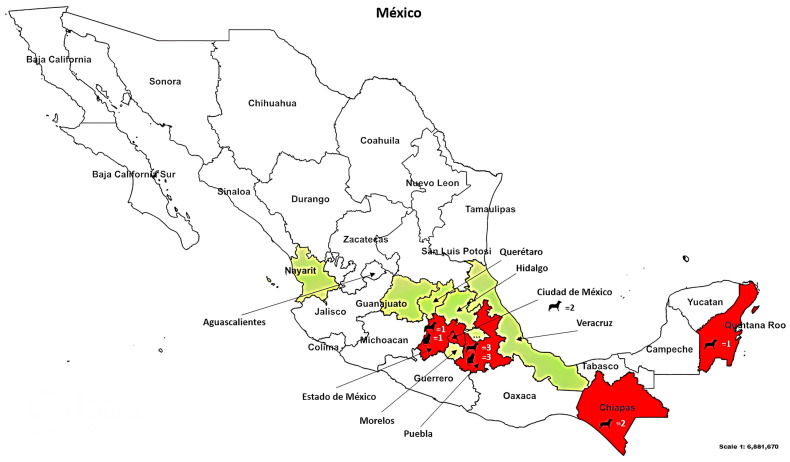
Map of Mexico with the SARS-CoV-2-positive dog and cat samples. The illustration represents the locations of dogs and cats samples; red color-coded states had positive samples, white states represent negative samples, while green states represent those where no samples were collected.

**Figure 2 pathogens-12-00835-f002:**
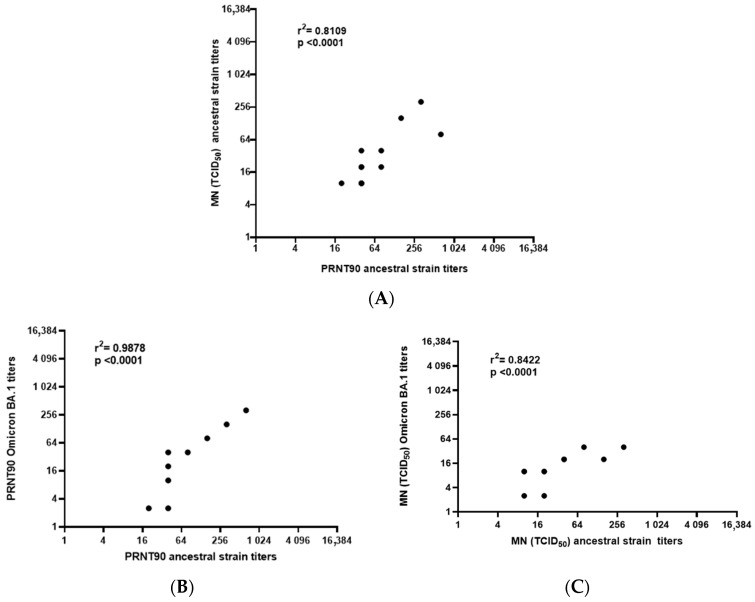
Pearson correlation between PRNT90 and MN. Pearson correlation was performed between PRNT90 and MN using the ancestral strain and positive samples from dogs and cats (**A**). Pearson correlation was performed between the results obtained with the Omicron subvariant BA.1 and ancestral strain, using the PRNT90 (**B**) and MN (**C**). In both cases, positive samples from dogs and cats were used. PRNT90: plaque reduction neutralization test; MN: microneutralization.

**Table 1 pathogens-12-00835-t001:** Dog and cat serum sample characteristics.

Source	Cats (n)	Dogs (n)	Total	State of Origin
Household	9	264	273	Coahuila, Chihuahua, Mexico City, Durango, Estado de México, Michoacán, Nuevo León, Puebla, Yucatán, and Jalisco.
Urban free-ranging	13	142	155	Chihuahua, México City, Guerrero, Estado de Mexico, Nuevo León, Puebla, Quintana Roo and Yucatán.
Shelter	6	118	124	Estado de Mexico, Mexico City, Puebla, and Yucatán.
Federal border inspection agencies	0	50	50	Aguas Calientes, Baja California Norte, Baja California Sur, Campeche, Colima, Chiapas, Chihuahua, Mexico City, Durango, Jalisco, Estado de Mexico, Nuevo León, Oaxaca, Quintana Roo, San Luis Potosí, Sinaloa, Tabasco, Tamaulipas, Yucatán, and Zacatecas.
Total	28	574	602	

**Table 2 pathogens-12-00835-t002:** Neutralizing antibody titers against SARS-CoV-2 in cats.

ID	Origin	State	Health History	PRNT90 * Ancestral (B.1.189)	MN ** Ancestral (B.1.189)	PRNT90 Omicron (BA.1)	MN Omicron (BA.1)
546	Household	Puebla	Healthy	1:320	1:320	1:160	1:40
547	Household	Puebla	Healthy	1:640	1:80	1:320	1:40
548	Household	Puebla	Healthy	1:160	1:160	1:80	1:20
476	Shelter	Estado de Mexico	Healthy	1:40	1:20	1:40	1:10

* PRNT90: plaque reduction neutralization test; ** MN, microneutralization assay.

**Table 3 pathogens-12-00835-t003:** Neutralizing antibody titers against SARS-CoV-2 in dogs.

ID	Source	State of Origin	Health History	PRNT90 * Ancestral (B.1.189)	MN ** Ancestral (B.1.189)	PRNT90 Omicron (BA.1)	MN Omicron (BA.1)
545	Household	Puebla	Healthy	1:40	1:10	negative	negative
549	Household	Puebla	Severe disease	1:20	1:10	negative	negative
539	Urban free-ranging	Mexico City	Healthy	1:80	1:20	1:40	1:10
538	Urban free-ranging	Puebla	Healthy	1:40	1:20	1:10	negative
483	Shelter	Estado de Mexico	Moderately ill	1:40	1:10	1:10	negative
540	Federal border inspection agencies	Chiapas	Healthy	1:80	1:40	1:40	1:20
1075	Federal border inspection agencies	Mexico City	Moderately ill	1:40	1:10	1:20	1:10
1092	Federal border inspection agencies	Quintana Roo	Moderately ill	1:40	1:40	1:40	1:20
1106	Federal border inspection agencies	Chiapas	Healthy	1:40	1:10	1:20	1:10

* PRNT90: plaque reduction neutralization test; ** MN, microneutralization assay.

## Data Availability

Data are available upon request.

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
