# Peer review of "Neutralizing Antibodies against the SARS-CoV-2 Ancestral Strain and Omicron BA.1 Subvariant in Dogs and Cats in Mexico"

_pathogens, 2023, doi:10.3390/pathogens12060835_

Round 1
Reviewer 1 Report (Previous Reviewer 2)
Th authors didn't make the required comments
Th authors didn't make the required comments
Author Response
Response to Reviewer 1 comments
First, we would like to thank the opportunity to submit a newly revised version of our manuscript.
In the first round of revision, three reviewers evaluated our manuscript. We are unsure if Reviewer 1 in this second round corresponds to Reviewer 1 in the first round. Considering that the only Reviewer with concerns about the "discarded serum samples" was reviewer 2 (in the first round), we concluded that Reviewer 2 in the first round corresponded to Reviewer 1 in this second revision.
Consequently, we will focus on these comments because the Reviewer commented, "Th authors didn't make the required comments." Probably our response did not satisfy the reviewer criteria, but we confirm that we answered all the comments. We have improved our responses to meet the Reviewer's exigencies.
Reviewer 2
Summary
This manuscript is an original article that evaluate the seroprevalence of neutralizing antibodies against the ancestral strain and Omicron BA.1 subvariants in dogs and cats from different regions of Mexico. I think there is definitely a need for this type of publication due to misleading information that is circulating about COVID-19; however, I have many concerns about this manuscript, that is require a major revision.
Thank you again for the opportunity to submit a newly revised version.
Comments
- Abstract
1- The abstract provides further information about the work's background and aim of work. It has to be paraphrased, to emphasise the study's findings and conclusion more.
We confirm that the abstract was previously modified as requested. In this new version, we made some additional changes.
2- Please arrange the keywords alphabetically.
We confirm that the keywords were arranged accordingly.
- Introduction
3- Line 39-40, please update the number of COVID-19 cases. It is not 4 million, it was exceeding 687 million cases.
We confirm that we updated the information in the previous version. In this new version, we updated the information to May 31.
4- Line 40-42, please update the number of COVID-19 cases in Mexico until May 2023.
We confirm that we updated the information in the previous version. In this new version, we updated the information to May 31.
5- Line 55-59, What is the reference of this paragraph?
We have included the references as suggested. In this new version, the change is in line 311.
6- Please provide more information about COVID-19 variants emergence.
We have improved the information as requested. Lines 52.63
7- Please specify the duration of sample collection in introduction aim of work.
We have included the requested information.
III. Materials and Methods
8- Please include the animal ethics statement.
We have included the requested information. Lines 99-102.
9- Not used certified and approved reference strains and reference serum in negative and positive control.
A: The viruses used in this study were sequenced and deposited in the GenBank. We do not use a reference positive and negative control, but we prepared negative and positive controls from non-infected and infected guinea pigs (please check section 2.7)
- Results
10- Please adjust the resolution and the quality of figure 1.
We have improved the resolution and quality of Figure 1.
11- Please add the total number of samples for rows and columns in table 1.
We have modified the tables as requested.
12- What does the authors means by discarded serum samples?
We have removed the discarded serum samples.
- Discussion
13- The discussion needs to be paraphrased, more information regarding the study and its findings should be added.
We have reorganized the discussion section as suggested.
- References
14- The references need major revision.
We have improved the format of references as suggested.
- Some references have DOIs, whereas others do not.
A: Done as suggested.
- Some references are missing page numbers, while others are missing an issue or volume number.
A: Done as suggested.
VII. General comments
15- The manuscript needs major English editing.
Done as suggested
Reviewer 2 Report (Previous Reviewer 3)
The authors have addressed most of the reviewer's concerns in the revised manuscript. I recommend this paper for acception.
Author Response
Thank you for your positive comments.
Round 2
Reviewer 1 Report (Previous Reviewer 2)
Accept in present form
Accept in present form
This manuscript is a resubmission of an earlier submission. The following is a list of the peer review reports and author responses from that submission.
Round 1
Reviewer 1 Report
In the present study, Freddy-Dehesa et al describe the occurrence of neutralizing antibodies against the SARS-CoV-2 in dogs and cats in Mexico.
For this, serum from 1052 dogs and 59 cats was collected from different regions of Mexico, and the presence of neutralizing antibodies (nAbs) was evaluated using a plaque reduction neutralization test (PRNT90) and microneutralization (MN). Among cats, 15.25% were positive, while 2.85% of dogs presented nAbs.
The study is of potential interest to the audience and good standards were followed for the investigation.
Specific comments:
- Third paragraph of the introduction (line 52 – 64) can be moved to discussion.
- MM (lines 87 – 88) remove or define “significant number” and “massive days”.
- MM (line 138) confirm the number of cells per well. 1.5 x 10^6/well seems to be a higher concentration than usual.
- MM (line 147) “four ten fold dil from 10 to 10^5” is 5 dil.
- MM (line 162) number of cells “1.5 x 10^6”. Please define if it was per well or plate.
- MM (line 173) “samples were considered positive when a serum dilution of at least 1:10 did not neutralize”.
- Map (Fig. 1) needs to be improved. Use a larger font size. If possible, add data about samples collected/tested in addition to positivity.
- Results (lines 209 - 210) “Titers against the ancestral strain in cats were higher than those against the Omicron BA.1”. This statement must be supported statically. Please add analysis.
- Tables 1 and 2. “Seroprevalence…” must be replaced by cats positive for NAbs against SARS-CoV-2.
- Results (lines 213 – 215) review and define each sample accordantly as cat or dog serum when describing IDs.
- Figure 2. Panel A – it is unclear if analyses were performed using data from ancestral or Omicron or both. Please clarify.
- Figure 2. Panel B – “using positive samples from dogs (n = 1052) and cats (n = 59)”. Is this number of positive or total samples tested?
- Discussion. Overinterpretation of the results should be avoided. The number of dog and cat samples and the diverse samples obtained from dogs limit comparison. The data presented in the present study is insufficient for the authors to say that cats are more susceptible than dogs. (line 270).
- Line 297. “In this
study, we found that cats had higher antibody titers than dogs, consistent with
previous reports [13, 20, 21, 35]. These percentages are similar…” Which
percentages are the authors referring to?
- Lines 306 and in the entire paragraph, the authors discuss cats' and dogs' reinfection based on the neutralizing activity of the guinea pig serum from animals immunized using inactivated virus and adjuvant. There needs to be more than this data to provide a conclusion. The paragraph can be deleted.
Reviewer 2 Report
Summary
This manuscript is an original article that evaluate the seroprevalence of neutralizing antibodies against the ancestral strain and Omicron BA.1 subvariants in dogs and cats from different regions of Mexico. I think there is definitely a need for this type of publication due to misleading information that is circulating about COVID-19; however, I have many concerns about this manuscript, that is require a major revision.
Comments
I. Abstract
1- The abstract provides further information about the work's background and aim of work. It has to be paraphrased, to emphasise the study's findings and conclusion more.
2- Please arrange the keywords alphabetically.
II. Introduction
3- Line 39-40, please update the number of COVID-19 cases. It is not 4 million, it was exceeding 687 million cases.
4- Line 40-42, please update the number of COVID-19 cases in Mexico until May 2023.
5- Line 55-59, What is the reference of this paragraph?
6- Please provide more information about COVID-19 variants emergence.
7- Please specify the duration of sample collection in introduction aim of work.
III. Materials and Methods
8- Please include the animal ethics statement.
9- Not used certified and approved reference strains and reference serum in negative and positive control.
IV. Results
10- Please adjust the resolution and the quality of figure 1.
11- Please add the total number of samples for rows and columns in table 1.
12- What does the authors means by discarded serum samples?
V. Discussion
13- The discussion needs to be paraphrased, more information regarding the study and its findings should be added.
VI. References
14- The references need major revision.
· Some references have DOIs, whereas others do not.
· Some references are missing page numbers, while others are missing an issue or volume number.
VII. General comments
15- The manuscript needs major English editing.
The manuscript needs major English editing.
Reviewer 3 Report
Review of Dehesa et al.
In this work, Dehesa et al collected 1111 samples of dog and cats from different regions in Mexico. They evaluated the serum neutralizing antibodies against SARS-CoV-2 ancestral strain and Omicron BA.1 by using PRNT90 and MN. 15.25% of cats and 2.85% of dogs showed nAbs against SARS-CoV-2 ancestral strain, nAbs positive percentage against SARS-CoV-2 Omicron BA.1 was the same in cats but with lower titers, while nAbs against SARS-CoV-2 Omicron BA.1 was lower (2.09%) in dogs. In general, cats showed higher frequency of nAbs compared to dogs, while the potency of nAbs was reduced against Omicron BA.1. Although the authors collected a large number of samples as noted by the reviewer, my enthusiasm for this paper for this journal is dampened by the limited nature of the findings and the inappropriate of experiment design.
Major issue
1. Significance. (a) Similar findings have been published before in other places; it is not surprising that the authors showed the same results in Mexico. (b) What is the significance of analyzing the nAbs levels in these animals? Is it known that dogs/cats are an important route of transmission to humans for SARS-CoV-2?
2. Rigor. (a) In this study, the authors totally collected 1111 samples from cats and dogs. But do the authors know when these animals were infected with SARS-CoV-2? In other words, how long after SARS-CoV-2 infection were samples taken? This is crucial since antibody titers/levels may vary a lot at different stages and this will affect the data and conclusions. The authors should be careful when stating their results. (b) Do the authors know which SARS-CoV-2 variant(s) were these animals infected? It is well known different variants will trigger different profiles of neutralizing antibodies. So, it is not appropriate to combine all the samples and perform comparison if they were not the same.
Minor comments
1. There are some typos/grammar mistakes in the manuscript, please go over the whole text and fix them.
2. Figure 1, can authors provide better quality figure? The resolution is too low. The font size is too small to read.
3. Table 2 and table 3, can the authors change the length of some boxes to make each sample fit in one row?
4. Figure legends, “Figure 1” is Italic, “Figure 2” is not. Please make them consistent.
See attached.